# From Alexnet to Transformers: Measuring the Non-linearity of Deep Neural Networks with Affine Optimal Transport

## Abstract

In the last decade, we have witnessed the introduction of several novel deep neural network (DNN) architectures exhibiting ever-increasing performance across diverse tasks. Explaining the upward trend of their performance, however, remains difficult as different DNN architectures of comparable depth and width – common factors associated with their expressive power – may exhibit a drastically different performance even when trained on the same dataset. In this paper, we introduce the concept of the non-linearity signature of DNN, the first theoretically sound solution for approximately measuring the non-linearity of deep neural networks. Built upon a score derived from closed-form optimal transport mappings, this signature provides a better understanding of the inner workings of a wide range of DNN architectures and learning paradigms, with a particular emphasis on the computer vision task. We provide extensive experimental results that highlight the practical usefulness of the proposed non-linearity signature and its potential for long-reaching implications.

## 1 Introduction

Deep neural networks (DNNs) are undoubtedly the most powerful AI models currently available [1, 2, 3, 4, 5]. Their performance on many tasks, including natural language processing (NLP) [6] and computer vision [7], is already on par or exceeds that of a human being. One of the reasons explaining such progress is of course the increasing computational resources [8, 9]. Another one is the endeavour for finding ever more efficient neural architectures pursued by researchers over the last decade. As of today, the transformer architecture [10] has firmly imposed itself as a number one choice for most, if not all, of the recent breakthroughs [11, 12, 13] in the machine learning and artificial intelligence fields.

**Limitations**  But why transformers are more capable than other architectures? Answering this question requires finding a meaningful measure to compare the different famous models over time gauging the trend of their intrinsic capacity. For such a comparison to be informative, it is particularly appropriate to consider the computer vision field that produced many of the landmark neural architectures improving upon each other over the years. Indeed, the decade-long revival of deep learning started with Alexnet's [14] architecture, the winner of the ImageNet Large Scale Visual Recognition Challenge [15] in 2012. By achieving a significant improvement over the traditional approaches, Alexnet was the first truly deep neural network to be trained on a dataset of such scale, suggesting that deeper models were likely to bring even more gains. In the following years, researchers proposed novel ways to train deeper models with hundreds of layers [16, 17, 18, 19] pushing the performance frontier even further. The AI research landscape then reached a turning

point with the proposal of transformers [10], starting their unprecedented dominance first in NLP and then in computer vision [20]. Surprisingly, transformers are not particularly deep, and the size of their landmark vision architecture is comparable to that of Alexnet, and this despite a significant performance gap between the two. Ultimately, this gap should be explained by the differences in the expressive power [21] of the two models: a term used to denote the ability of a DNN to approximate functions of a certain complexity. Unfortunately, the existing theoretical results related to this either associate higher expressive power with depth [22, 23, 24] or width [25, 26, 27, 28] falling short in comparing different families of architectures. This, in turn, limits our ability to understand what underpins the achieved progress and what challenges and limitations still exist in the field, guiding future research efforts.

**Contributions** We argue that quantifying the non-linearity of a DNN may be what we were missing so far to understand the evolution of the deep learning models at a more fine-grained level. To verify this hypothesis in practice, we put forward the following contributions:

1. We propose a first theoretically sound measure, called the affinity score, that estimates the non-linearity of a given (activation) function using optimal transport (OT) theory. We use the proposed affinity score to introduce the concept of the non-linearity signature of DNNs defined as a set of affinity scores of all its activation functions.

2. We compare non-linearity signatures of a wide range of popular DNNs used in computer vision: from Alexnet to vision transformers (ViT) and their more recent variations. Through this, we clearly illustrate the disruptive patterns in the evolution of the deep learning field.

3. We demonstrate that non-linearity signature can be predictive of DNNs performance and used to meaningfully identify the family of approaches to which a given DNN belongs. We further show that the non-linearity signature is unique as it doesn't correlate strongly with other potential candidates used for this task.

The rest of the paper is organized as follows. We start by presenting the relevant background knowledge on OT in Section 2. Then, we introduce the affinity score together with its different theoretical properties in Section 3. Section 4 presents experimental evaluations on a wide range of popular convolutional neural networks. Finally, we conclude in Section 5.

## 2 Background

**Optimal Transport** Let $(X, d)$ be a metric space equipped with a lower semi-continuous *cost function* $c : X \times X \to \mathbb{R}_{\geq 0}$, e.g the Euclidean distance $c(x, y) = \|x - y\|$. Then, the Kantorovich formulation of the OT problem between two probability measures $\mu, \nu \in \mathcal{P}(X)$ is given by

$$\text{OT}_c(\mu, \nu) = \min_{\gamma \in \text{ADM}(\mu, \nu)} \mathbb{E}_\gamma[c], \tag{1}$$

where $\text{ADM}(\mu, \nu)$ is the set of joint probabilities with marginals $\mu$ and $\nu$, and $\mathbb{E}_\nu[f]$ denotes the expected value of $f$ under $\nu$. The optimal $\gamma$ minimizing equation 1 is called the *OT plan*. Denote by $\mathcal{L}(X)$ the law of a random variable $X$. Then, the OT problem extends to random variables $X, Y$ and we write $\text{OT}_c(X, Y)$ meaning $\text{OT}_c(\mathcal{L}(X), \mathcal{L}(Y))$.

Assuming that either of the considered measures is *absolutely continuous*, then the Kantorovich problem is equivalent to the *Monge problem*

$$\text{OT}_c(\mu, \nu) = \min_{T : T_\# \mu = \nu} \mathbb{E}_{X \sim \mu}[c(X, T(X))], \tag{2}$$

where the unique minimizing $T$ is called the *OT map*, and $T_\# \mu$ denotes the *push-forward measure*, which is equivalent to the *law* of $T(X)$, where $X \sim \mu$.

**Wasserstein distance** Let $X$ be a random variable over $\mathbb{R}^d$ satisfying $\mathbb{E}[\|X - x_0\|^2] < \infty$ for some $x_0 \in \mathbb{R}^d$, and thus for any $x \in \mathbb{R}^d$. We denote this class of random variables by $\mathcal{P}_2(\mathbb{R}^d)$. Then, the 2-Wasserstein distance $W_2$ between $X, Y \in \mathcal{P}_2(\mathbb{R}^d)$ is defined as

$$W_2(X, Y) = \text{OT}_{\|x-y\|^2}(X, Y)^{\frac{1}{2}}. \tag{3}$$

We now proceed to the presentation of our main contribution.

# 3 Non-linearity signature of deep neural networks

Among all non-linear operations introduced into DNNs in the last several decades, activation functions remain the only structural piece that they all inevitably share. Without non-linear activation functions, most of DNNs, no matter how deep, reduce to a linear function unable to learn complex patterns. Activation functions were also early identified [29, 30, 31, 32] as a key to making even a shallow network capable of approximating any function, however complex it may be, to arbitrary precision.

We thus build our study on the following intuition: if activation functions play in important role in making DNNs non-linear, then measuring their degree of non-linearity can provide us with an approximation of the DNN's non-linearity itself. To implement this intuition in practice, however, we first need to find a way to measure the non-linearity of an activation function. Surprisingly, there is no widely accepted measure for this, neither in the field of mathematics nor in the field of computer science. To fill this gap, we will use the OT theory to develop a so-called *affinity score* below.

## 3.1 Affinity score

**Identifiability**  We consider the pre-activation signal $X$ of an activation function within a neural network, and the post-activation signal $\sigma(X)$ denoted by $Y$ as input and output random variables. Our first step to build the affinity score then is to ensure that we can identify when $\sigma$ is linear with respect to (wrt) $X$ (for instance, when an otherwise non-linear activation is *locally linear* at the support of $X$). To show that such an identifiability condition can be satisfied with OT, we first recall the following classic result from the literature characterizing the OT maps.

**Theorem 3.1** ([33]). *Let $X \in \mathcal{P}_2(\mathbb{R}^d)$, $T(x) = \nabla\phi(x)$ for a convex function $\phi$ with $T(X) \in \mathcal{P}_2(\mathbb{R}^d)$. Then, $T$ is the unique optimal OT map between $\mu$ and $T_{\#}\mu$.*

Using this theorem about the uniqueness of OT maps expressed as gradients of convex functions, we can prove the following result (all proofs can be found in the Appendix C):

**Corollary 3.2.** *Without loss of generality, let $X, Y \in \mathcal{P}_2(\mathbb{R}^d)$ be centered, and let $Y = \sigma(X) = TX$, where $T$ is a positive definite linear transformation. Then, $T$ is the OT map from $X$ to $Y$.*

Whenever the activation function $\sigma$ is linear, the solution to the OT problem $T$ exactly reproduces it.

**Characterization**  We now seek to understand whether $T$ can be characterized more explicitly. For this, we prove the following theorem stating that $T$ can be computed in closed-form using the normal approximations of $X$ and $Y$.

**Theorem 3.3.** *Let $X, Y \in \mathcal{P}_2(\mathbb{R}^d)$ be centered and $Y = TX$ for a positive definite matrix $T$. Let $N_X \sim \mathcal{N}(\mu(X), \Sigma(X))$ and $N_Y \sim \mathcal{N}(\mu(Y), \Sigma(Y))$ be their normal approximations where $\mu$ and $\Sigma$ denote mean and covariance, respectively. Then, $W_2(N_X, N_Y) = W_2(X, Y)$ and $T = T_{\mathrm{aff}}$, where $T_{\mathrm{aff}}$ is the OT map between $N_X$ and $N_Y$ and can be calculated in closed-form*

$$T_{\mathrm{aff}}(x) = Ax + b, \quad A = \Sigma(Y)^{\frac{1}{2}} \left( \Sigma(Y)^{\frac{1}{2}} \Sigma(X) \Sigma(Y)^{\frac{1}{2}} \right)^{-\frac{1}{2}} \Sigma(Y)^{\frac{1}{2}},$$
$$b = \mu(Y) - A\mu(X). \tag{4}$$

**Upper bound**  When the activation $\sigma$ is non-linear wrt $X$, the affine OT mapping $T_{\mathrm{aff}}(X)$ will deviate from the true activation outputs $Y$. One important step toward quantifying this deviation is given by the famous Gelbrich bound, formalized by means of the following theorem:

**Theorem 3.4** (Gelbrich bound [34]). *Let $X, Y \in \mathcal{P}_2(\mathbb{R}^d)$ and let $N_X, N_Y$ be their normal approximations. Then, $W_2(N_X, N_Y) \leq W_2(X, Y)$.*

This upper bound provides a first intuition of why OT can be a great tool for measuring non-linearity: the cost of the affine map solving the OT problem on the left-hand side increases when the map becomes non-linear. We now upper bound the difference between $W_2(N_X, N_Y)$ and $W_2(X, Y)$, two quantities that coincide *only* when $\sigma$ is linear.

**Proposition 3.5.** *Let $X, Y \in \mathcal{P}_2(\mathbb{R}^d)$ and $N_X, N_Y$ be their normal approximations. Then,*

1. $|W_2(N_X, N_Y) - W_2(X, Y)| \leq \dfrac{2\,\mathrm{Tr}\left[(\Sigma(X)\Sigma(Y))^{\frac{1}{2}}\right]}{\sqrt{\mathrm{Tr}[\Sigma(X)] + \mathrm{Tr}[\Sigma(Y)]}}.$

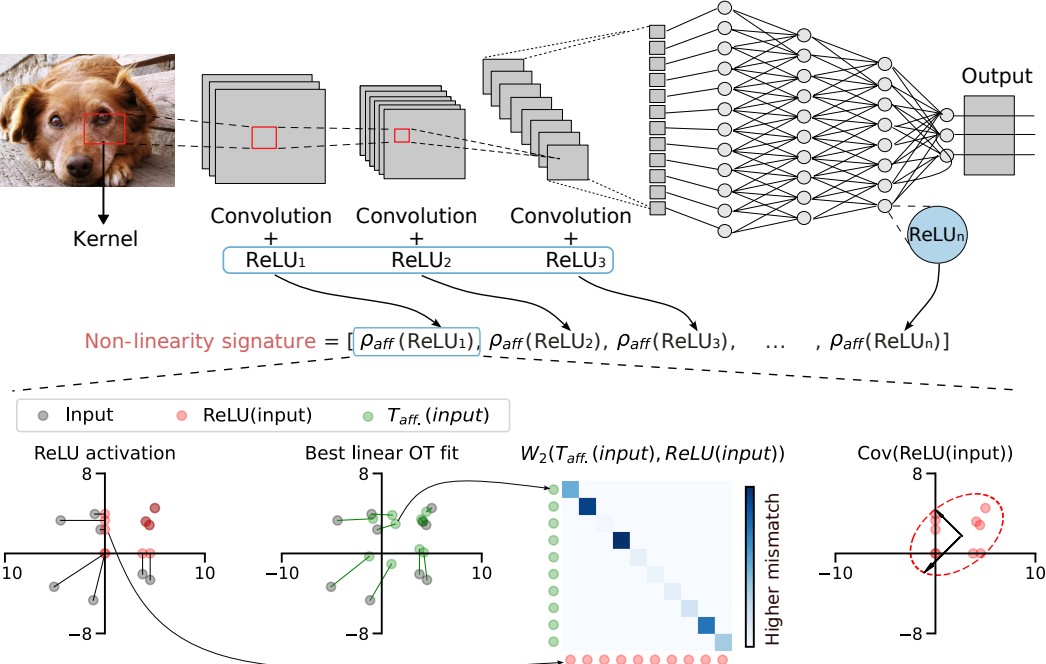

Figure 1: Illustration of how the non-linearity of a given neural network is measured. (**Top**) The non-linearity signature of a DNN is a collection of affinity scores calculated for each activation function spread across its hidden layers. (**Bottom**) The affinity score is calculated based on 3 main steps. First, given an input (grey) and an output (red) of an activation function (*left*), we estimate the best affine OT fit $T_{\text{aff}}(X)$ (green) transporting the input to the output (*middle-left*). Second, we measure the mismatch between the two by summing the transportation costs (*middle-right*) to obtain the Wasserstein distance $W_2(T_{\text{aff}}X, Y)$. Finally, this distance is normalized with the magnitudes of variance (arrows in the rightmost plot) of the output data based on its covariance matrix.

2. For $T_{\text{aff}}$ as in (4), $W_2(T_{\text{aff}}X, Y) \leq \sqrt{2 \operatorname{Tr}\left[\Sigma(Y)\right]}$.

To have a more informative non-linearity measure, we now need to normalize the non-negative Wasserstein distance $W_2(T_{\text{aff}}X, Y)$ to an interpretable interval of $[0, 1]$. The bound given in Proposition 3.5 lets us define the following *affinity score*

$$\rho_{\text{aff}}(X, \sigma(X)) = 1 - \frac{W_2(T_{\text{aff}}X, \sigma(X))}{\sqrt{2 \operatorname{Tr}[\Sigma(\sigma(X))]}}. \tag{5}$$

The proposed affinity score quantifies how far a given activation $\sigma$ is from an affine transformation. It is equal to 1 for any input for which the activation function is linear, and 0 when it is maximally non-linear, i.e., when $T_{\text{aff}}X$ and $\sigma(X)$ are independent random variables.

**Remark 3.6.** *One may wonder whether a simpler alternative to the affinity score can be to use, instead of $T_{\text{aff}}$, a mapping $T_W(x) = Wx$ defined as a solution of a linear regression problem $\min_W ||Y - WX||_F^2$. Then, one can use the coefficient of determination ($R^2$ score) to measure how well $T_W$ fits the observed data. This approach, however, has two drawbacks. First, following the famous Gauss-Markov theorem, $T_W$ is an optimal linear (linear in $Y$) estimator. On the contrary, $T_{\text{aff}}$ is a globally optimal non-linear mapping aligning $X$ and $Y$. Second, $R^2$ compares the fit of $T_W$ with that of a mapping outputting $\mu(Y)$ for any value of $X$. This is contrary to $\rho_{\text{aff}}$ that compares how well $T_{\text{aff}}$ fits the data wrt to the worst possible cost incurred by $T_{\text{aff}}$ as quantified in Proposition 3.5. This gives us a bounded score, i.e. $\rho_{\text{aff}} \in [0, 1]$, whereas $R^2$ is not lower bounded, i.e. $R^2 \in [-\infty, 1]$. We confirm experimentally in Section 4 that the two coefficients do not correlate consistently across the studied DNNs suggesting that $R^2$ is a poor proxy to $\rho_{\text{aff}}$.*

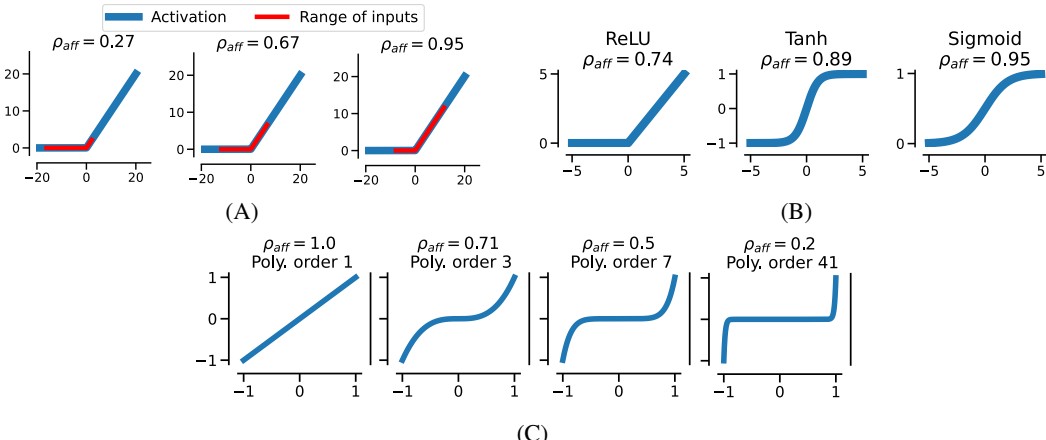

Figure 2: **(A)** Non-linearity of ReLU depends on the range of input values (*red*); **(B)** ReLU, Tanh, and Sigmoid exhibit different degrees of non-linearity for the same input; **(C)** Affinity score captures the increasing non-linearity of polynomials of different degrees.

### 3.2 Non-linearity signature

We now turn our attention to the definition of a non-linearity signature of deep neural networks. We define a neural network N as a composition of layers $F_i$ where each layer $F_i$ is a function taking as input a tensor $X_i \in \mathbb{R}^{h_i \times w_i \times c_i}$ (for instance, an image of size $224 \times 224 \times 3$ for $i = 1$) and outputting a tensor $Y_i \in \mathbb{R}^{h_{i+1} \times w_{i+1} \times c_{i+1}}$ used as an input of the following layer $F_{i+1}$. This defines $N = F_L \odot ... \odot F_i ... \odot F_1 = \bigodot_{k=1,...,L} F_k$ where $\odot$ stands for a composition.

We now present the definition of a non-linearity signature of a network N. Below, we abuse the compositional structure of $F_i$ and see it as an ordered sequence of functions.

**Definition 3.1.** *Let $N = \bigodot_{k=1,...,L} F_k$ be a neural network. Define by $\mathcal{A}$ a finite set of common activation functions such that $\mathcal{A} := \{\sigma | \sigma : \mathbb{R}^{h \times w \times c} \to \mathbb{R}^{h \times w \times c}\}$. Let $r$ be a pooling operation such that $r : \mathbb{R}^{h \times w \times c} \to \mathbb{R}^c$. Then, the non-linearity signature of N given an input X is defined as follows:*

$$\rho_{\mathrm{aff}}(N; X) = \{\rho_{\mathrm{aff}}(r(X_i), \sigma(r(X_i))), \quad \forall \sigma \in F_i \cap \mathcal{A}, \quad i = \{1, \ldots, L\}\}.$$

Non-linearity signature, illustrated in Figure 1, associates to each network N a vector of affinity scores calculated over the inputs and outputs of all activation functions encountered across its layers.

**What makes an activation function non-linear?** We now want to understand the mechanism behind achieving a lower or higher non-linearity with a given (activation) function. This will explain what the different values of the affinity scores stand for when defining the non-linearity signature of a DNN. In Figure 2(A), we show how the ReLU function [35], defined element-wise as $\mathrm{ReLU}(x) = \max(0, x)$, achieves its varying degree of non-linearity. Interestingly, this degree depends only on the range of the input values. Second, in Figure 2(B) we also show how the shape of activation functions impacts their non-linearity for a fixed input: surprisingly, piece-wise linear ReLU function is more non-linear than $\mathrm{Sigmoid}(x) = 1/(e^{-x} + 1)$ [36] or $\mathrm{Tanh}(x) = (e^{-x} - e^x)/(e^{-x} + e^x)$. Similar observations also apply to compare polynomials of varying degrees (Figure 2(C)). We refer the reader to Appendix D for more visualizations of the affinity score of popular activation functions.

### 3.3 Related work

**Layer-wise similarity analysis of DNNs** A line of work that can be distantly related to our main proposal is that of quantifying the similarity of the hidden layers of the DNNs as proposed [37] and [38] (see [39] for a complete survey of the subsequent works). [37] extracts activation patterns of the hidden layers in the DNNs and use CCA on the singular vectors extracted from them to measure how similar the two layers are. Their analysis brings many interesting insights regarding the learning dynamics of the different convnets, although they do not discuss the non-linearity propagation in the

convnets, nor do they propose a way to measure it. [38] proposed to use a normalized Frobenius inner product between kernel matrices calculated on the extracted activations of the hidden layers and argued that such a similarity measure is more meaningful than that proposed by [37].

**Impact of activation functions**   [40] provides the most comprehensive survey on the activation functions used in DNNs. Their work briefly discusses the non-linearity of the different activation functions suggesting that piecewise linear activation functions with more linear components are more non-linear (e.g., ReLU vs. ReLU6). [41] show theoretically that smooth versions of ReLU allow for more efficient information propagation in DNNs with a positive impact on their performance. Our work provides a first extensive comparison of all popular activation functions; we also show that smooth version of ReLU exhibit wider regions of high non-linearity (see Appendix D).

**Non-linearity measure**   The only work similar to ours in spirit is the paper by [42] proposing the non-linearity coefficient in order to predict the train and test error of DNNs. Their coefficient is defined as a square root of the Jacobian of the neural network calculated wrt its input, multiplied by the covariance matrix of the Jacobian, and normalized by the covariance matrix of the input. The presence of the Jacobian in it calls for the differentiability assumption making its application to most of the neural networks with ReLU non-linearity impossible as is. The authors didn't provide any implementation of their coefficient and we were not able to find any other study reporting the reproduced results from this work.

## 4   Experimental evaluations

We consider computer vision models trained and evaluated on the same Imagenet dataset with 1,000 output categories (Imagenet-1K) publicly available at [43]. The non-linearity signatures of different studied models presented in the paper is calculated by passing batches of size 512 through the pre-trained models for the entirety of the Imagenet-1K validation set (see Appendix H for more datasets) with a total of 50,000 images. We include the following landmark architectures in our study: Alexnet [14], four VGG models [16], Googlenet [44], Inception v3 [17], five Resnet models [18], four Densenet models [19], four MNASNet models [45], four EfficientNet models [46], five ViT models, three Swin transformer [47] and four Convnext models [48]. We include MNASNet and EfficientNet models as prominent representatives of the neural architecture search approach [49]. Such models are expected to explicitly maximize the accuracy for a given computational budget. Swin transformer and Convnext models are introduced as ViTs with traditional computer vision priors. Their presence will be useful to better grasp how such priors impact ViTs. We refer the reader to Appendix E for more practical details.

**History of deep vision models at a glance**   We give a general outlook of the developments in computer vision over the last decade when seen through the lens of their non-linearity. In Figure 3 we present the minimum, median, and maximum values of the affinity scores calculated for the considered neural networks (see Appendix F for raw non-linearity signatures). We immediately see that until the arrival of transformers, the trend of the landmark models was to decrease their non-linearity, rather than to increase it. On a more fine-grained level, we note that pure convolution architectures such as Alexnet (2012) and VGGs (2014) exhibit a very low spread of the affinity score values. This trend changes with the arrival of the inception module first used in Googlenet (2014): the latter includes activation functions that extend the range of the non-linearity on both ends of the spectrum. Importantly, we can see that the trend toward increasing the maximum and average non-linearity of the neural networks has continued for almost the whole decade. Even more surprisingly, EfficientNet models (2019), trained through neural architecture search, have strong negative skewness toward higher linearity, although they were state-of-the-art in their time. The second surprising finding comes with the arrival of ViTs (2020): they break the trend and leverage the non-linearity of their hidden activation functions becoming more or more non-linear with the varying size of the patches (see Appendix F for a more detailed comparison with raw signatures). This trend remains valid also for Swin transformers (2021), although introducing the computer vision priors into them makes their non-linearity signature look more similar to pure convolutional networks from the early 2010s, such as Alexnet and VGGs. Finally, we observe that the non-linearity signature of a modern Convnext architecture (2022), designed as a convnet for 2020s using the best practices of Swin transformers, further confirms this observation.

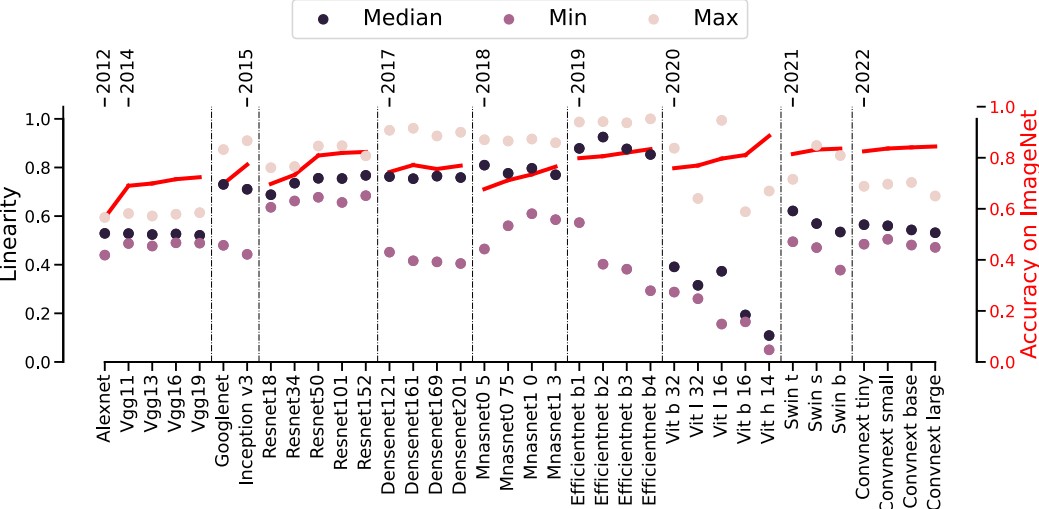

Figure 3: Median, minimum, and maximum values of non-linearity signatures of the different architectures spanning a decade (2012-2022) of computer vision research. We observe a clear trend toward the increase of the spread and the maximum values of the linearity in neural networks lasting until the arrival of transformers in 2020. ViTs have a distinct pattern of maximizing the non-linearity of their activation functions. Swin transformers and Convnext models retain this property from them while remaining close to the pure convolutional networks.

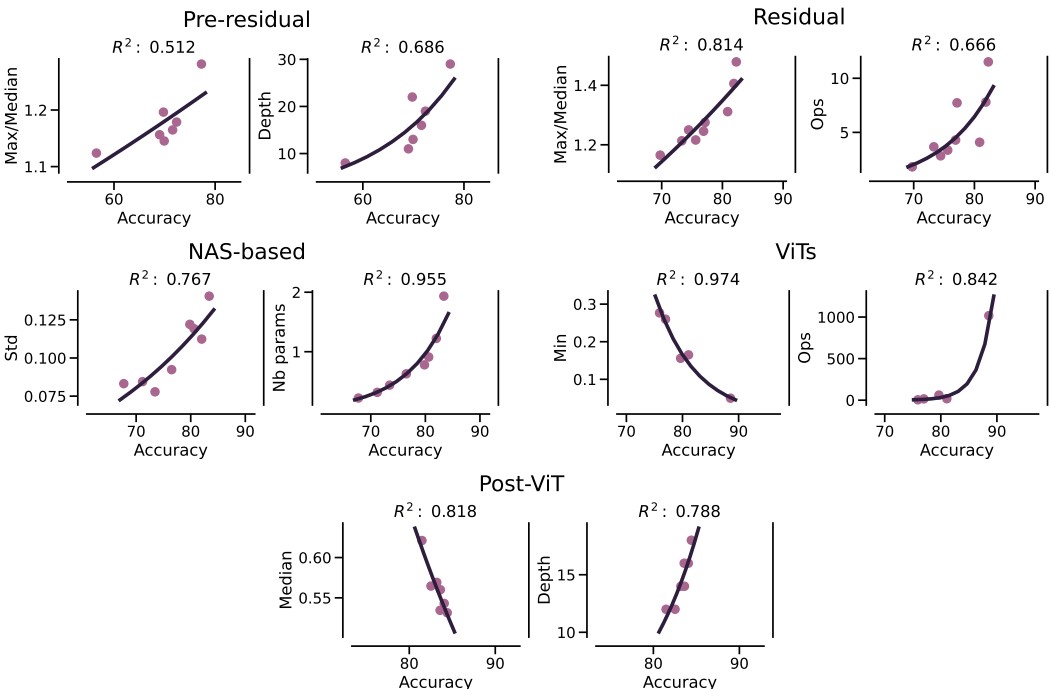

Figure 4: Best found dependency between the different statistics extracted from the non-linearity signatures of the DNN families and their respective Imagenet-1K accuracy. The results are compared in terms of the $R^2$ score against the most precise of the other common DNN characteristics such as depth, size, and the GFLOPS.

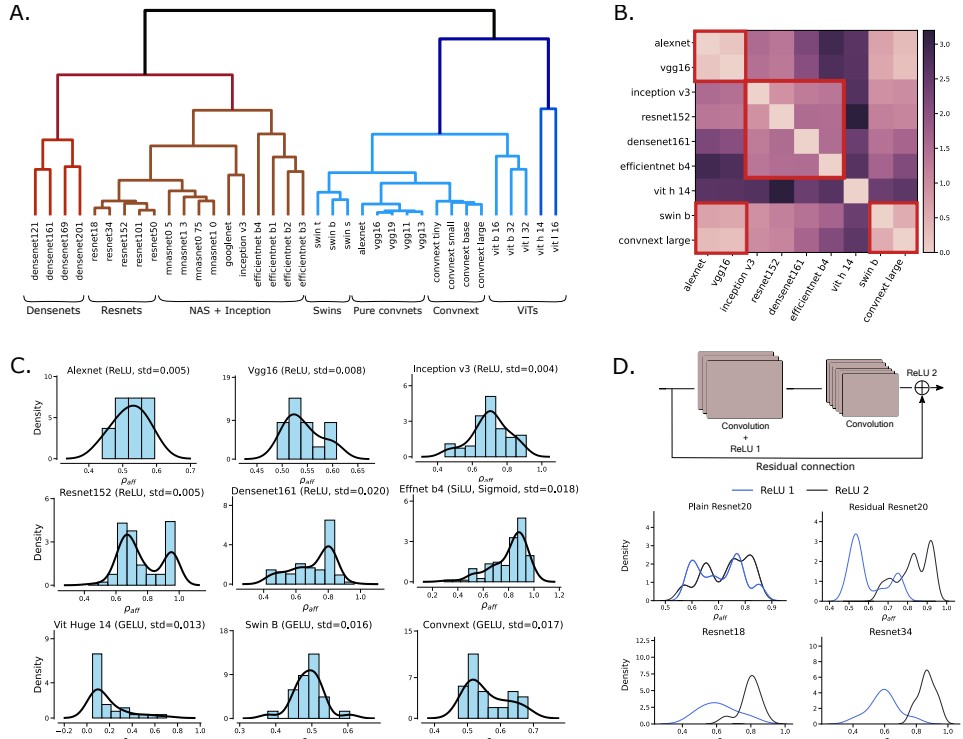

Figure 5: Comparing the different families of the neural architectures based on their non-linearity signatures. **(A)** Hierarchical clustering of all DNNs considered in our study revealing meaningful clusters with close architectural characteristics; **(B)** 9 representative architectures from all studied families and the similarities between them. Note how the similarities between early convnets and other models is decreasing with time until computer vision priors are introduced into Swin transformers in 2021; **(C)** Distributions of affinity scores in each network. Most models expand the non-linearity ranges of their activation functions compared to early convnets. ViTs are dominated by highly non-linear activation functions, Resnets have a bimodal distribution, Densenets, and EfficientNets have a diametrically skewed distribution compared to ViTs. **(D)** Comparing the same convnet with 20 layers when trained with (Residual Resnet20) and without (Plain Resnet20) residual connections (top row). Residual connections introduce a clear trend toward a bimodal distribution of affinity scores; the same effect is observed for Resnet18 and Resnet34 (bottom row).

**Closer look at accuracy/non-linearity trade-off** Different families of vision models leverage differ-
ent characteristics of their internal non-linearity to achieve better performance. To better understand
this phenomenon, we now turn our attention to a more detailed analysis of the accuracy/non-linearity
trade-off by looking for a statistic extracted from their non-linearity signatures that is the most predic-
tive of their accuracy as measured by the $R^2$ score. Additionally, we also want to understand whether
the non-linearity of DNNs can explain their performance better than the traditional characteristics
such as the number of parameters, the number of giga floating point operations per second (GFLOPS),
and the depth. From the results presented in Figure 4, we observe the following. First, the information
extracted from the non-linearity signatures often correlates more with the final accuracy, than the
usual DNN characteristics. This is the case for Residual networks (ResNets and DenseNets), ViTs,
and vision models influenced by transformers (Post-ViT). Unsurprisingly, for models based on neural
architecture search (NAS-based, i.e. EfficientNets and MNASNets) the number of parameters is
the most informative metric as they are specifically designed to reach the highest accuracy with the
increasing model size and compute. For Pre-residual pure convolutional models (Alexnet, VGGs,
Googlenet, and Inception), the spread of the non-linearity explains the accuracy increase similarly to
depth. Second, we observe that all models preceding ViTs were implicitly optimizing the spread of
their affinity score values to achieve better performance. After the arrival of the transformers, the
observed trend is to increase either the median or the minimum values of the non-linearity. This
suggests a fundamental shift in the implicit bias that the transformers carry.

Table 1: Pearson correlations between the non-linearity signature and other metrics, for all the architectures evaluated in this study. The highest absolute value in each group is reported in **bold**.

| Models | CKA | NORM | SPARSITY | ENTROPY | $R^2$ |
|---|---|---|---|---|---|
| VGGs | $0.0 \pm 0.05$ | $-0.67 \pm 0.06$ | $-0.18 \pm 0.03$ | $\mathbf{-0.90 \pm 0.04}$ | $-0.21 \pm 0.06$ |
| ResNets | $0.53 \pm 0.04$ | $-0.41 \pm 0.19$ | $\mathbf{-0.68 \pm 0.02}$ | $-0.38 \pm 0.12$ | $-0.48 \pm 0.24$ |
| DenseNets | $0.88 \pm 0.02$ | $-0.76 \pm 0.02$ | $\mathbf{-0.89 \pm 0.02}$ | $-0.66 \pm 0.03$ | $0.85 \pm 0.04$ |
| MNASNets | $\mathbf{0.67 \pm 0.11}$ | $-0.54 \pm 0.14$ | $-0.63 \pm 0.07$ | $-0.55 \pm 0.16$ | $0.45 \pm 0.17$ |
| EfficientNets | $\mathbf{0.42 \pm 0.10}$ | $-0.16 \pm 0.22$ | $-0.17 \pm 0.23$ | $-0.16 \pm 0.14$ | $0.21 \pm 0.12$ |
| ViTs | $-0.22 \pm 0.40$ | $\mathbf{-0.67 \pm 0.20}$ | $-0.09 \pm 0.56$ | $0.17 \pm 0.25$ | $-0.10 \pm 0.34$ |
| Swins | $-0.15 \pm 0.13$ | $\mathbf{-0.53 \pm 0.10}$ | $-0.26 \pm 0.17$ | $0.06 \pm 0.35$ | $-0.13 \pm 0.13$ |
| Convnexts | $0.69 \pm 0.08$ | $0.21 \pm 0.15$ | $0.23 \pm 0.16$ | $0.02 \pm 0.09$ | $\mathbf{0.79 \pm 0.05}$ |
| Average | $0.33 \pm 0.45$ | $\mathbf{-0.44 \pm 0.34}$ | $-0.32 \pm 0.42$ | $-0.31 \pm 0.39$ | $0.14 \pm 0.49$ |

**Distinct signature for every architecture**     Non-linearity signature correctly identifies the different families of neural architectures. To show this, we perform hierarchical clustering using pairwise dynamic time warping (DTW) distances [50] between the non-linearity signatures of the models from Figure 3. The results in Figure 5 (A), as well as the pairwise distance matrix between a representative of each studied family in Figure 5 (B) (see Appendix G for the full matrix), show that we correctly cluster all similar models together, both within their respective families (such as the different variations of the same architecture) and across them (such as the cluster of Swin and pure convolution models). Additionally, we highlight the individual affinity scores' distributions of representative models in Figure 5 (C). Finally, we highlight the exact effect of residual connections proposed in 2016 and used ever since by every benchmark model in Figure 5 (D). It reveals vividly that residual connections make the distribution of the affinity scores bimodal with one such mode centered around highly linear activation functions. This confirms in a principled way that residual connections indeed tend to enable the learning of the identity function just as suggested in the seminal work that proposed them [18]. Non-linearity signatures can also be applied to meaningfully identify training methods, such as popular nowadays self-supervised approaches, for a fixed architecture (see Appendix I).

**Uniqueness of the affinity score**   No other metric extracted from the activation functions of the considered networks exhibits a strong consistent correlation with the non-linearity signature. To validate this claim, we compare in Table 1 the Pearson correlation between the non-linearity signature and several other metrics comparing the inputs and the outputs of the activation functions. We can see that for different models the non-linearity correlates with different metrics suggesting that it captures the information that other metrics fail to capture consistently across all architectures. This becomes even more apparent when analyzing the individual correlation values (in Appendix G). Overall, the proposed affinity score and the non-linearity signatures derived from it offer a unique perspective on the developments in the ML field.

## 5   Discussions

We proposed the first sound approach to measure non-linearity of activation functions in neural networks and defined their non-linearity signature based on it. We further used non-linearity signatures to provide a meaningful overview of the evolution of neural architectures proposed over the last decade with clear interpretable patterns. We showed that until the arrival of transformers, the trend in DNNs was to decrease their non-linearity, rather than to increase it. Vision transformers changed this pattern drastically. We also showcased that our measure is unique, as no other metric correlates strongly with it across all architectures.

In the future, our work can be applied to study the non-linearity of the LLM models to better understand the effect of different architectural choices in them. On a higher level, our approach can also be used to identify new disruptive neural architectures by identifying those of them that leverage different internal non-linearity characteristics to obtain better performance. This capacity of identifying novel technologies is even more crucial in the age of very large models where experimenting with the building blocks of the optimized backbone comes at a very high cost.

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

## A  Broader Impacts

This paper presents work whose goal is to advance the field of Machine Learning and better understand the underlying behavior of Deep Neural Networks architectures. There are many potential societal consequences of our work, none which we feel must be specifically highlighted here.

## B  Limitations

An important assumption of Theorem 3.3, is that the activation function that we want to analyze through $\rho_{\text{aff}}$ needs to be a positive definite transformation of the inputs. Fortunately, this is the case for activation functions, that we consider in this paper. Finally, we note that despite the strong correlation between the statistics extracted from the non-linearity signatures for certain DNNs' architectures, we are yet to show that explicitly optimizing affinity scores through backpropagation can have an actionable impact on DNNs performance or its other properties, such as robustness or transferability.

## C  Proofs of main theoretical results

In this section, we provide proofs of the main theoretical results from the paper.

**Corollary 3.2.** Without loss of generality, let $X, Y \in \mathcal{P}_2(\mathbb{R}^d)$ be centered, and such that $Y = TX$, where $T$ is a positive semi-definite linear transformation. Then, $T$ is the OT map from $X$ to $Y$.

*Proof.* We first proof that we can consider centered distributions without loss of generality. To this end, we note that

$$W_2^2(X, Y) = W_2^2(X - \mathbb{E}[X], Y - \mathbb{E}[Y]) + \|\mathbb{E}[X] - \mathbb{E}[Y]\|^2, \tag{6}$$

implying that splitting the 2-Wasserstein distance into two independent terms concerning the $L^2$ distance between the means and the 2-Wasserstein distance between the centered measures.

Furthermore, if we have an OT map $T'$ between $X - \mathbb{E}[X]$ and $Y - \mathbb{E}[Y]$, then

$$T(x) = T'(x - \mathbb{E}[X]) + \mathbb{E}[Y], \tag{7}$$

is the OT map between $X$ and $Y$.

To prove the statement of the Corollary, we now need to apply Theorem 3.1 to the convex $\phi(x) = x^T T x$, where $T$ is positive semi-definite. □

**Theorem 3.3.** Let $X, Y \in \mathcal{P}_2(\mathbb{R}^d)$ be centered and $Y = TX$ for a positive definite matrix $T$. Let $N_X \sim \mathcal{N}(\mu(X), \Sigma(X))$ and $N_Y \sim \mathcal{N}(\mu(Y), \Sigma(Y))$ be their normal approximations where $\mu$ and $\Sigma$ denote mean and covariance, respectively. Then, $W_2(N_X, N_Y) = W_2(X, Y)$ and $T = T_{\text{aff}}$, where $T_{\text{aff}}$ is the OT map between $N_X$ and $N_Y$ and can be calculated in closed-form

$$T_{\text{aff}}(x) = Ax + b, \quad A = \Sigma(Y)^{\frac{1}{2}} \left( \Sigma(Y)^{\frac{1}{2}} \Sigma(X) \Sigma(Y)^{\frac{1}{2}} \right)^{-\frac{1}{2}} \Sigma(Y)^{\frac{1}{2}},$$
$$b = \mu(Y) - A\mu(X). \tag{8}$$

*Proof.* Corollary 3.2 states that $T$ is an OT map, and

$$\Sigma(TN_X) = T\Sigma(X)T = \Sigma(Y).$$

Therefore, $TN_X = N_Y$, and by Theorem 3.1, $T$ is the OT map between $N_X$ and $N_Y$. Finally, we compute

$$W_2^2(N_X, N_Y) = \text{Tr}[\Sigma(X)] + \text{Tr}[T\Sigma(X)T] - 2\,\text{Tr}[T^{\frac{1}{2}}\Sigma(X)T^{\frac{1}{2}}]$$
$$= \underset{T:T(X)=Y}{\arg\min} \ \mathbb{E}_X[\|X - T(X)\|^2]$$
$$= W_2^2(X, Y).$$

□

**Proposition 3.5.** Let $X, Y \in \mathcal{P}_2(\mathbb{R}^d)$ and $N_X, N_Y$ be their normal approximations. Then,

1. $|W_2(N_X, N_Y) - W_2(X, Y)| \leq \dfrac{2 \operatorname{Tr}\left[(\Sigma(X)\Sigma(Y))^{\frac{1}{2}}\right]}{\sqrt{\operatorname{Tr}[\Sigma(X)] + \operatorname{Tr}[\Sigma(Y)]}}$.

2. For $T_{\text{aff}}$ as in (4), $W_2(T_{\text{aff}} X, Y) \leq \sqrt{2} \operatorname{Tr}[\Sigma(Y)]^{\frac{1}{2}}$.

*Proof.* By Theorem 3.4, we have $W_2(N_X, N_Y) \leq W_2(X, Y)$. On the other hand,

$$
\begin{aligned}
W_2^2(X, Y) &= \min_{\gamma \in \text{ADM}(X,Y)} \int_{\mathbb{R}^d \times \mathbb{R}^d} \|x - y\|^2 d\gamma(x, y) \\
&\leq \int_{\mathbb{R}^d \times \mathbb{R}^d} \left( \|x\|^2 + \|y\|^2 \right) d\gamma(x, y) \\
&= \operatorname{Tr}[\Sigma(X)] + \operatorname{Tr}[\Sigma(Y)].
\end{aligned}
$$

Combining the above inequalities, we get

$$
|W_2(N_X, N_Y) - W_2(X, Y)| \leq \left| \sqrt{\operatorname{Tr}[\Sigma(X)] + \operatorname{Tr}[\Sigma(Y)]} - W_2(N_X, N_Y) \right|.
$$

Let $a = \operatorname{Tr}[\Sigma(X)] + \operatorname{Tr}[\Sigma(Y)]$, and so $W_2^2(N_X, N_Y) = a - b$, where $b = 2 \operatorname{Tr}\left[(\Sigma(X)\Sigma(Y))^{\frac{1}{2}}\right]$.
Then the RHS of can be written as

$$
\left| \sqrt{a} - \sqrt{a - b} \right| = \frac{|a - (a - b)|}{\sqrt{a} + \sqrt{a - b}} \leq \frac{b}{\sqrt{a}},
$$

where the inequality follows from positivity of $W_2(N_X, N_Y) = \sqrt{a - b}$. Letting $X = T_{\text{aff}} X$ in the obtained bound gives 2). $\qquad \square$

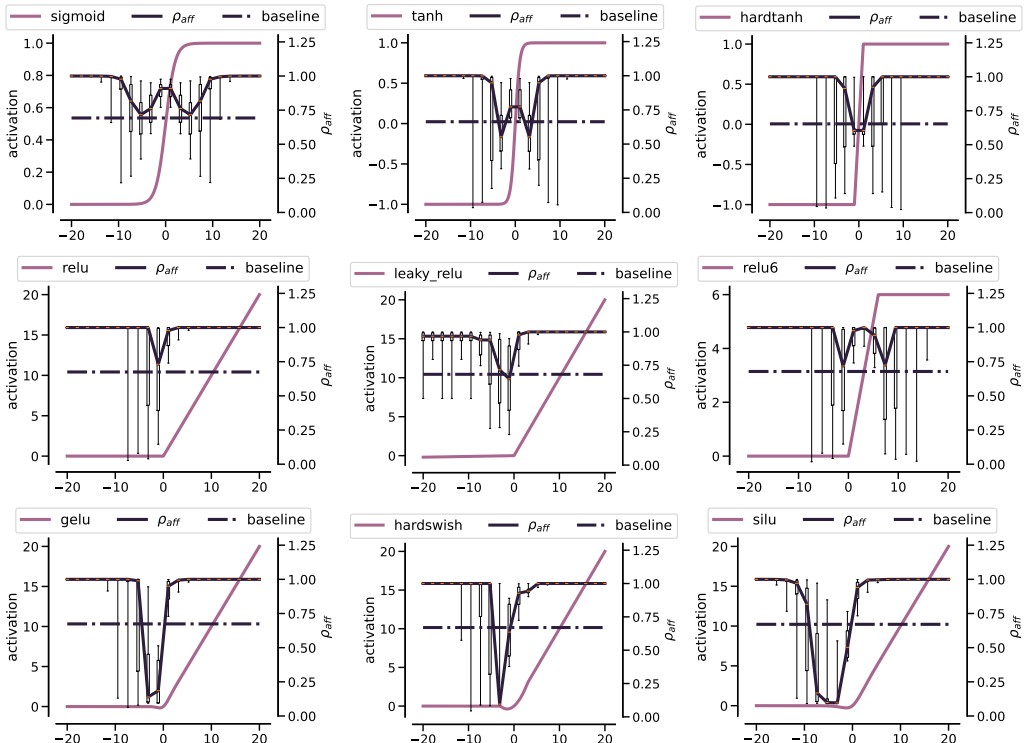

Figure 6: Median affinity scores of Sigmoid, ReLU, GELU, ReLU6, LeakyReLU with a default value of slope, Tanh, HardTanh, SiLU, and HardSwish obtained across random draws from Gaussian distribution with a sliding mean and varying stds used as their input. Whiskers of boxplots show the whole range of values obtained for each mean across all stds. The baseline value is the affinity score obtained for a sample covering the whole interval. The ranges and extreme values of each activation function over its subdomain are indicative of its non-linearity limits.

## D Affinity scores of other popular activation functions

Many works aimed to improve the way how the non-linearity – represented by activation functions – can be defined in DNNs. As an example, a recent survey on the commonly used activation functions in deep neural networks [40] identifies over 40 activation functions with first references to sigmoid dating back to the seminal paper [36] published in late 80s. The fashion for activation functions used in deep neural networks evolved over the years in a substantial way, just as the neural architectures themselves. Saturating activations, such as sigmoid and hyperbolic tan, inspired by computational neuroscience were a number one choice up until the arrival of rectifier linear unit (ReLU) in 2010. After being the workhorse of many famous models over the years, the arrival of transformers popularized Gaussian Error Linear Unit (GELU) which is now commonly used in many large language models including GPTs.

We illustrate in Figure 6 the affinity scores obtained after a single pass of the data through the following activation functions: Sigmoid, ReLU [51], GELU [52], ReLU6 [53], LeakyReLU [54] with a default value of the slope, Tanh, HardTanh, SiLU [55], and HardSwish [56]. As the non-linearity of activation functions depends on the domain of their input, we fix 20 points in their domain equally spread in $[-20, 20]$ interval. We use these points as means $\{m_i\}_{i=1}^{20}$ of Gaussian distributions from which we sample 1000 points in $\mathbb{R}^{300}$ with standard deviation (std) $\sigma$ taking values in $[2, 1, 0.5, 0.25, 0.1, 0.01]$. Each sample denoted by $X_{m_i}^{\sigma_j}$ is then passed through the activation function act $\in \{\text{sigmoid}, \text{ReLU}, \text{GELU}\}$ to obtain $\rho_{\text{aff}}^{m_i, \sigma_j} := \rho_{\text{aff}}(X_{m_i}^{\sigma_j}, \text{act}(X_{m_i}^{\sigma_j}))$. Larger std values make it more likely to draw samples that are closer to the region where the studied activation functions become non-linear. We present the obtained results in Figure S2 where each of 20 boxplots showcases median($\rho_{\text{aff}}^{m_i, \sigma_\cdot}$) values with 50% confidence intervals and whiskers covering the whole range of obtained values across all $\sigma_j$.

This plot allows us to derive several important conclusions. We observe that each activation function can be characterized by 1) the lowest values of its non-linearity obtained for some subdomain of the considered interval and 2) the width of the interval in which it maintains its non-linearity. We note that in terms of 1) both GELU and ReLU may attain affinity scores that are close to 0, which is not the case for Sigmoid. For 2), we observe that the non-linearity of Sigmoid and GELU is maintained in a wide range, while for ReLU it is rather narrow. We can also see a distinct pattern of more modern activation functions, such as SiLU and HardSwish having a stronger non-linearity pattern in large subdomains. We also note that despite having a shape similar to Sigmoid, Tanh may allow for much lower affinity scores. Finally, the variations of ReLU seem to have a very similar shape with LeakyReLU being on average more linear than ReLU and ReLU6.

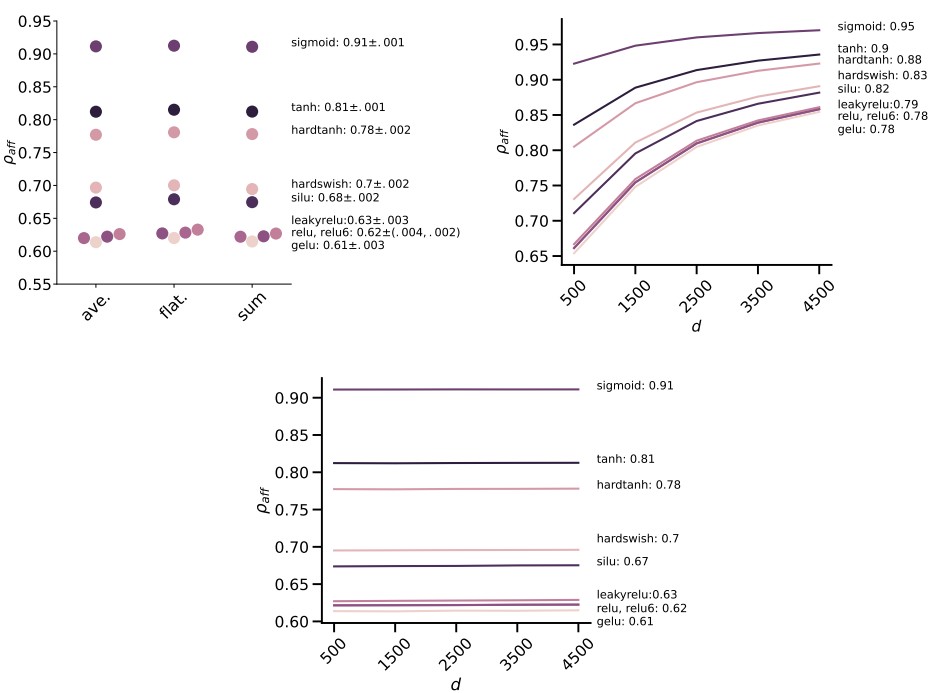

Figure 7: **(Top left)** Affinity score is robust to the dimensionality reduction both when using averaging and summation over the spatial dimensions; **(Top right)** When $d > n$, sample covariance matrix estimation leads to a lack of robustness in the estimation of the affinity score; **(Bottom)** Shrinkage of the covariance matrix leads to constant values of the affinity scores with increasing $d$.

## E Implementation details

**Dimensionality reduction**    Manipulating 4-order tensors is computationally prohibitive and thus we need to find an appropriate lossless function $r$ to facilitate this task. One possible choice for $r$ may be a vectorization operator that flattens each tensor into a vector. In practice, however, such flattening still leads to very high-dimensional data representations. In our work, we propose to use averaging over the spatial dimensions to get a suitable representation of the manipulated tensors. In Figure 7 (left), we show that the affinity score is robust wrt such an averaging scheme and maintains the same values as its flattened counterpart.

**Computational considerations**    The non-linearity signature requires calculating the affinity score over "wide" matrices. Indeed, after the reduction step is applied to a batch of $n$ tensors of size $h \times w \times c$, we end up with matrices of size $n \times c$ where $n$ may be much smaller than $c$. This is also the case when input tensors are 2D when the batch size is smaller than the dimensionality of the embedding space. To obtain a well-defined estimate of the covariance matrix in this case, we use a known tool from the statistics literature called Ledoit-Wolfe shrinkage [57]. In Figure 7 (right), we show that shrinkage allows us to obtain a stable estimate of the affinity scores that remain constant in all regimes.

**Robustness to batch size and different seeds**    In this section, we highlight the robustness of the non-linearity signature with respect to the batch size and the random seed used for training. To this end, we concentrate on VGG16 architecture and CIFAR10 dataset to avoid costly Imagenet retraining. In Figure 8, we present the obtained result where the batch size was varied between 128 and 1024 with an increment of 128 (left plot) and when VGG16 model was retrained with seeds varying from 1 to 9 (right plot). The obtained results show that the affinity score is robust to these parameters suggesting that the obtained results are not subject to a strong stochasticity.

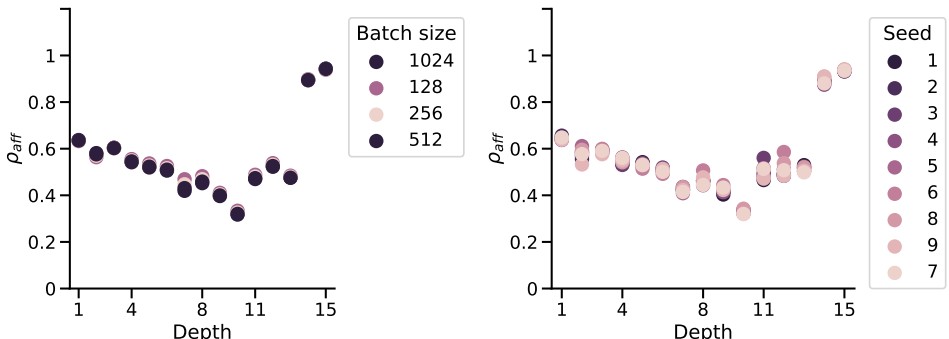

Figure 8: Non-linearity signature of VGG16 on CIFAR10 with a varying batch size (left) and when retrained from 9 different random seeds (right).

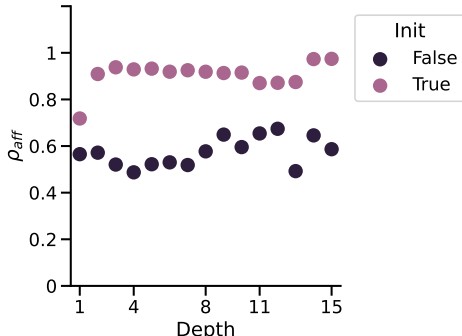

Figure 9: Non-linearity signatures of VGG16 on CIFAR10 in the beginning and end of training on Imagenet.

**Impact of training** Finally, we also show how a non-linearity signature of a VGG16 model looks like at the beginning and in the end of training on Imagenet. We extract its non-linearity signature at initialization when making a feedforward pass over the whole CIFAR10 dataset and compare it to the non-linearity signature obtained in the end. In Figure 9, we can see that at initialization the network's non-linearity signature is increasing, reaching almost a perfectly linear pattern in the last layers. Training the network enhances the non-linearity in a non-monotone way. Importantly, it also highlights that the non-linearity signature is capturing information from the training process.

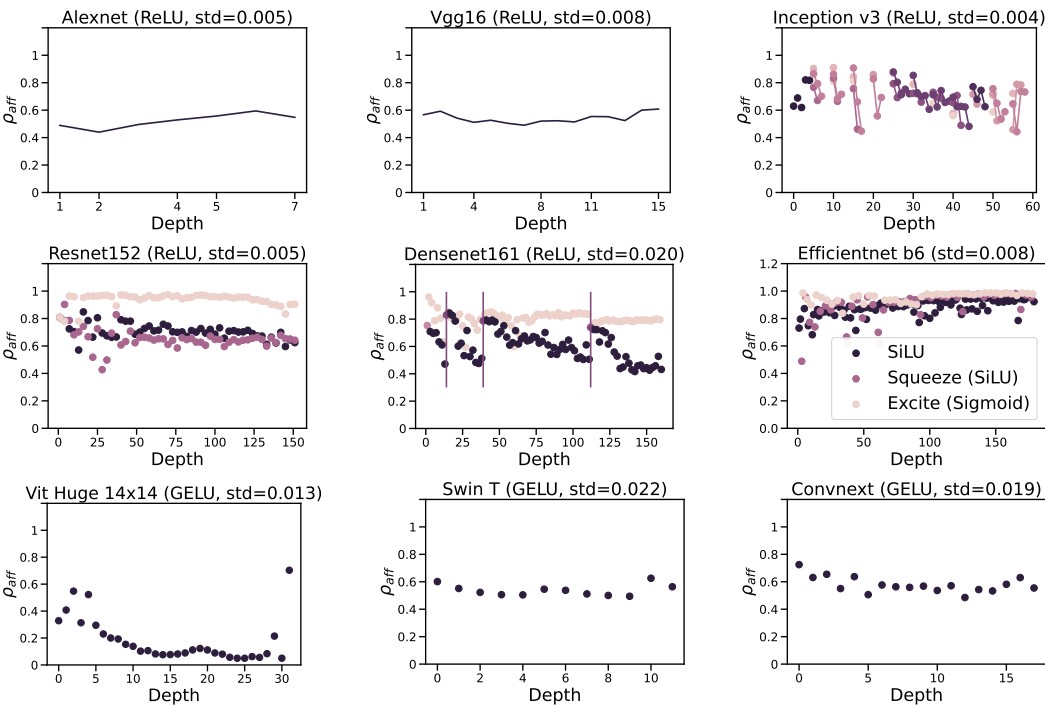

Figure 10: Raw non-linearity signatures of popular DNN architectures, plotted as affinity scores over the depth throughout the network.

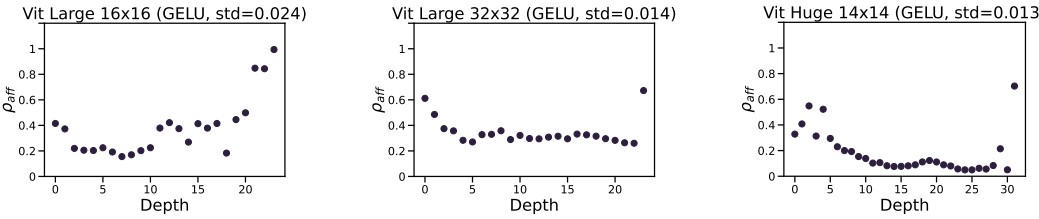

Figure 11: ViTs: Large ViT with 16x16 and 32x32 patch sizes and Huge ViT.

## F Raw signatures

In Figure 10, we portray the raw non-linearity signatures of several representative networks studied in the main paper. We use different color codes for distinct activation functions appearing repeatedly in the considered architecture (for instance, every first ReLU in a residual block of a Resnet). We also indicate the mean standard deviation of the affinity scores over batches in the title.

We see that the non-linearities across ReLU activations in all of Alexnet's 8 layers remain stable. Its successor, VGG network, reveals tiny, yet observable, variations in the non-linearity propagation with increasing depth and, slightly lower overall non-linearity values. We attribute this to the decreased size of the convolutional filters (3x3 vs. 7x7). The Googlenet architecture was the first model to consider learning features at different scales in parallel within the so-called inception modules. This add more variability as affinity scores of activation in Googlenet vary between 0.6 and 0.9. Despite being almost 20 times smaller than VGG16, the accuracy of Googlenet on Imagenet remains comparable, suggesting that increasing and varying the linearity is a way to have high accuracy with a limited computational complexity compared to predecessors. This finding is further confirmed with Inception v3 that pushed the spread of the affinity score toward being more linear in some hidden layers. When comparing this behavior with Alexnet, we note just how far we are from it. Resnets achieve the same spread of values of the non-linearity but in a different, and arguably, simpler way. Indeed, the activation after the skip connection exhibits affinity scores close to 1, while the activations in the hidden layers remain much lower. Densenet, that connect each layer to all previous layers and

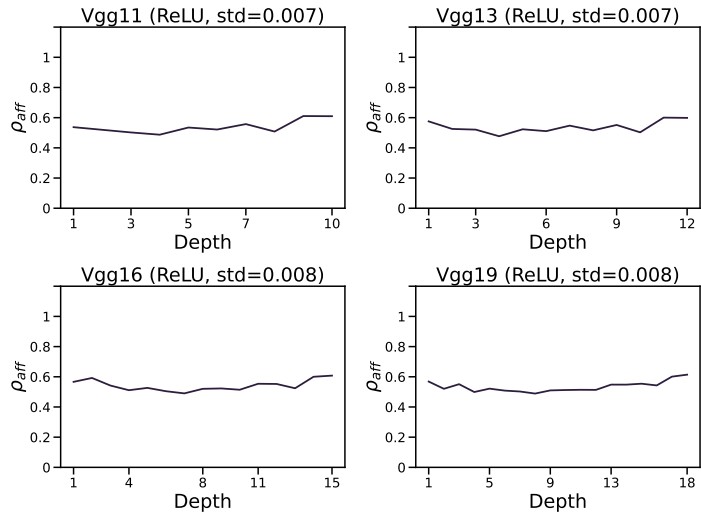

Figure 12: Impact of depth on the non-linearity signature of VGGs.

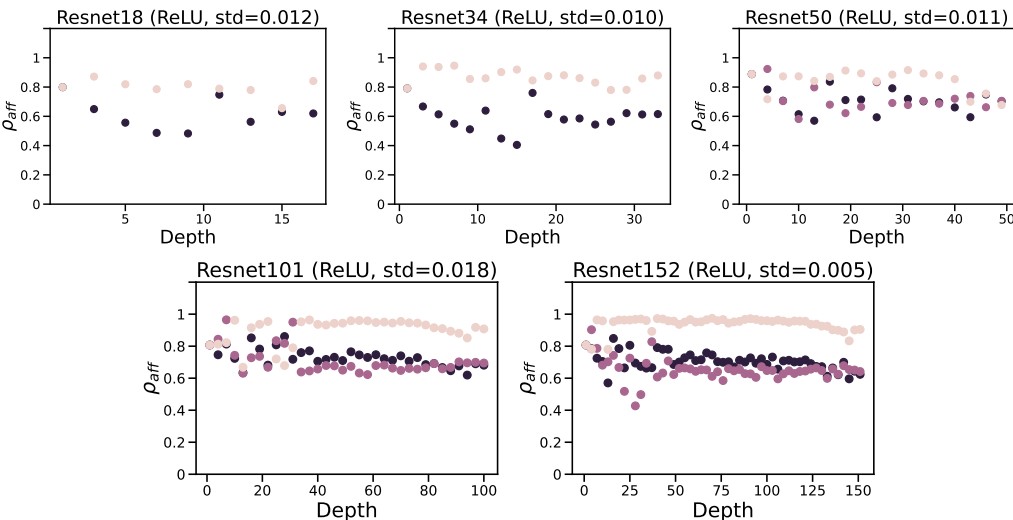

Figure 13: Impact of depth on the non-linearity signature of Resnets.

not just to the one that precedes it, is slightly more non-linear than Resnet152, although the two bear a striking similarity: they both have an activation function that maintains the non-linearity low with increasing depth. Additionally, transition layers in Densenet act as linearizers and allow it to reset the non-linearity propagation in the network by reducing the feature map size. ViTs (Large with 16x16 and 32x32 patch sizes, and Huge with 14x14 patches) are all highly non-linear models to the degree yet unseen. Interestingly, as seen in Figure 11 the patch size affects the non-linearity propagation in a non-trivial way: for 16x16 size a model is more non-linear in the early layers, while gradually becoming more and more linear later, while 32x32 patch size leads to a plateau in the hidden layers of MLP blocks, with a steep change toward linearity only in the final layer. We hypothesize that attention modules in ViT act as a focusing lens and output the embeddings in the domain where the activation function is the most non-linear.

Finally, we explore the role of increasing depth for VGG and Resnet architectures. We consider VGG11, VGG13, VGG16 and VGG19 models in the first case, and Resnet18, Resnet34, Resnet50, Resnet101 and Resnet152. The results are presented in Figure 12 and Figure 13 for VGGs and Resnets, respectively. Interestingly, VGGs do not change their non-linearity signature with increasing depth. In the case of Resnets, we can see that the separation between more linear post-residual activations becomes more distinct and approaches 1 for deeper networks.

Table 2: Pearson correlations between the affinity score and other metrics, for all the architectures evaluated in this study. We see that no other metric can reliably provide the same information as the proposed non-linearity signature across different neural architectures.

| Model | CKA | Norm | Sparsity | Entropy | $R^2$ |
|---|---|---|---|---|---|
| alexnet | -0.75 | **-0.86** | 0.14 | -0.80 | -0.41 |
| vgg11 | -0.07 | -0.76 | -0.15 | **-0.95** | -0.27 |
| vgg13 | 0.08 | -0.66 | -0.23 | **-0.93** | -0.26 |
| vgg16 | 0.01 | -0.63 | -0.19 | **-0.88** | -0.17 |
| vgg19 | -0.01 | -0.62 | -0.15 | **-0.86** | -0.14 |
| googlenet | 0.74 | -0.60 | **-0.83** | -0.49 | 0.73 |
| inception v3 | 0.69 | -0.66 | **-0.75** | -0.45 | 0.35 |
| resnet18 | 0.59 | -0.17 | **-0.67** | -0.30 | -0.44 |
| resnet34 | 0.48 | -0.18 | **-0.65** | -0.19 | -0.08 |
| resnet50 | 0.56 | -0.60 | -0.71 | -0.50 | **-0.78** |
| resnet101 | 0.51 | -0.57 | **-0.70** | -0.51 | -0.64 |
| resnet152 | 0.52 | -0.51 | **-0.68** | -0.42 | -0.48 |
| densenet121 | 0.84 | -0.75 | **-0.87** | -0.62 | 0.82 |
| densenet161 | **0.87** | -0.74 | **-0.87** | -0.67 | 0.81 |
| densenet169 | **0.87** | -0.74 | **-0.87** | -0.67 | 0.81 |
| densenet201 | 0.89 | -0.75 | **-0.91** | -0.67 | 0.90 |
| efficientnet b1 | 0.35 | **-0.41** | -0.39 | 0.01 | 0.03 |
| efficientnet b2 | **0.49** | -0.02 | -0.44 | -0.06 | 0.34 |
| efficientnet b3 | **0.32** | -0.12 | -0.18 | -0.13 | 0.18 |
| efficientnet b4 | 0.30 | **-0.51** | -0.29 | -0.44 | 0.11 |
| vit b 32 | 0.47 | -0.31 | -0.29 | 0.39 | **0.51** |
| vit l 32 | -0.14 | **-0.61** | -0.47 | -0.02 | -0.06 |
| vit b 16 | -0.27 | **-0.71** | 0.04 | 0.39 | -0.22 |
| vit l 16 | -0.39 | **-0.89** | -0.66 | -0.23 | -0.24 |
| vit h 14 | -0.77 | -0.83 | **0.92** | 0.31 | -0.49 |
| swin t | -0.12 | -0.39 | -0.02 | **-0.42** | -0.06 |
| swin s | -0.003 | **-0.61** | -0.31 | 0.18 | -0.03 |
| swin b | -0.32 | **-0.59** | -0.43 | 0.42 | -0.32 |
| convnext tiny | 0.77 | -0.01 | -0.04 | 0.09 | **0.80** |
| convnext small | 0.57 | 0.22 | 0.25 | 0.13 | **0.72** |
| convnext base | 0.67 | 0.41 | 0.35 | -0.03 | **0.82** |
| convnext large | 0.75 | 0.23 | 0.35 | -0.10 | **0.84** |
| Average | $0.31 \pm 0.45$ | **$-0.44 \pm 0.35$** | $-0.31 \pm 0.43$ | $-0.29 \pm 0.39$ | $0.13 \pm 0.50$ |

## G  Detailed comparisons between architectures

We consider the following metrics as 1) the linear CKA [38] commonly used to assess the similarity of neural representations, the average change in 2) SPARSITY and 3) ENTROPY before and after the application of the activation function as well as the 4) Frobenius NORM between the input and output of the activation functions, and the 5) $R^2$ score between the linear model fitted on the input and the output of the activation function. We present in Table 2, the detailed values of Pearson correlations obtained for each architecture and all the metrics considered in this study. In Figure 14, we show the full matrix of pairwise DTW distances [50] obtained between architectures, then used to obtain the clustering presented in the main text.

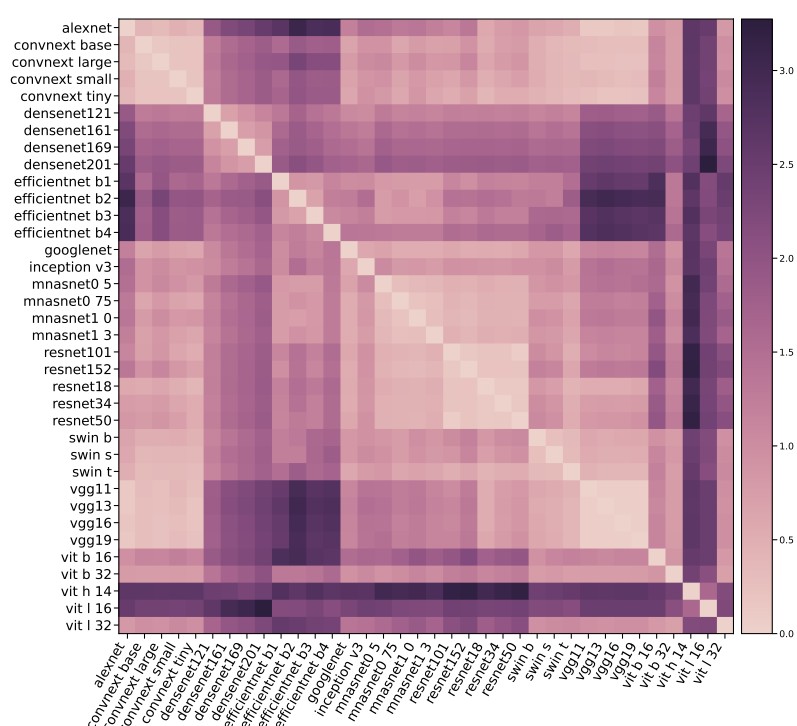

Figure 14: Full matrix of DTW distances between non-linearity signatures.

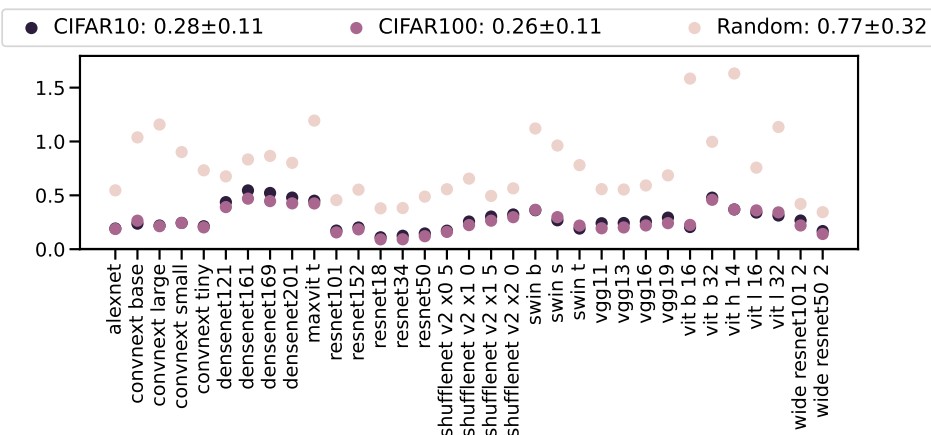

Figure 15: Deviation in terms of the Euclidean distance of the non-linearity signature obtained on CIFAR10, CIFAR100, and Random datasets from the non-linearity signature of the Imagenet dataset.

## H    Results on more datasets

Below, we compare the results obtained on CIFAR10, CIFAR100 datasets as well as when the random data tensors are passed through the network. As the number of plots for all chosen 33 models on these datasets will not allow for a meaningful visual analysis, we rather plot the differences – in terms of the DTW distance – between the non-linearity signature of the model on Imagenet dataset with respect to three other datasets. We present the obtained results in Figure 15.

We can see that the overall deviation for CIFAR10 and CIFAR100 remains lower than for Random dataset suggesting that these datasets are semantically closer to Imagenet.

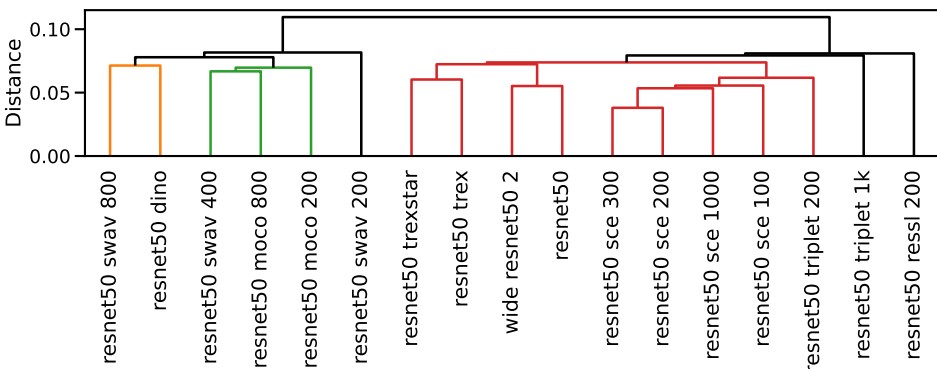

Figure 16: Hierarchical clustering of supervised and self-supervised pre-trained Resnet50 using the DTW distances between their non-linearity signatures.

Table 3: Robustness of the different criteria when considering the same architectures pre-trained for different tasks. Affinity score achieves the lowest standard deviation suggesting that it is capable of correctly identifying the architecture even when it was trained differently.

| Criterion | Mean $\pm$ std |
|---|---|
| $\rho_{\text{aff}}$ | 0.76$\pm$**0.04** |
| Linear CKA | 0.90$\pm$0.07 |
| Norm | 448.56$\pm$404.61 |
| Sparsity | 0.56$\pm$0.16 |
| Entropy | 0.39$\pm$0.46 |

# I   Results for self-supervised methods

In this section, we show that the non-linearity signature of a network remains almost unchanged when considering other pertaining methodologies such as for instance, self-supervised ones. To this end, we use 17 Resnet50 architecture pre-trained on Imagenet within the next 3 families of learning approaches:

1. SwAV [58], DINO [59], and MoCo [60] that belong to the family of contrastive learning methods with prototypes;

2. Resnet50 [18], Wide Resnet50 [61], TRex, and TRex* [62] that are supervised learning approaches;

3. SCE [63], Truncated Triplet [64], and ReSSL [65] that perform contrastive learning using relational information.

From the dendrogram presented in Figure 16, we can observe that the DTW distances between the non-linearity signatures of all the learning methodologies described above allow us to correctly cluster them into meaningful groups. This is rather striking as the DTW distances between the different instances of the Resnet50 model are rather small in magnitude suggesting that the affinity scores still retain the fact that it is the same model being trained in many different ways.

While providing a fine-grained clustering of different pre-trained models for a given fixed architecture, the average affinity scores over batches remain surprisingly concentrated as shown in Table 3. This hints at the fact that the non-linearity signature is characteristic of architecture but can also be subtly multi-faceted when it comes to its different variations.

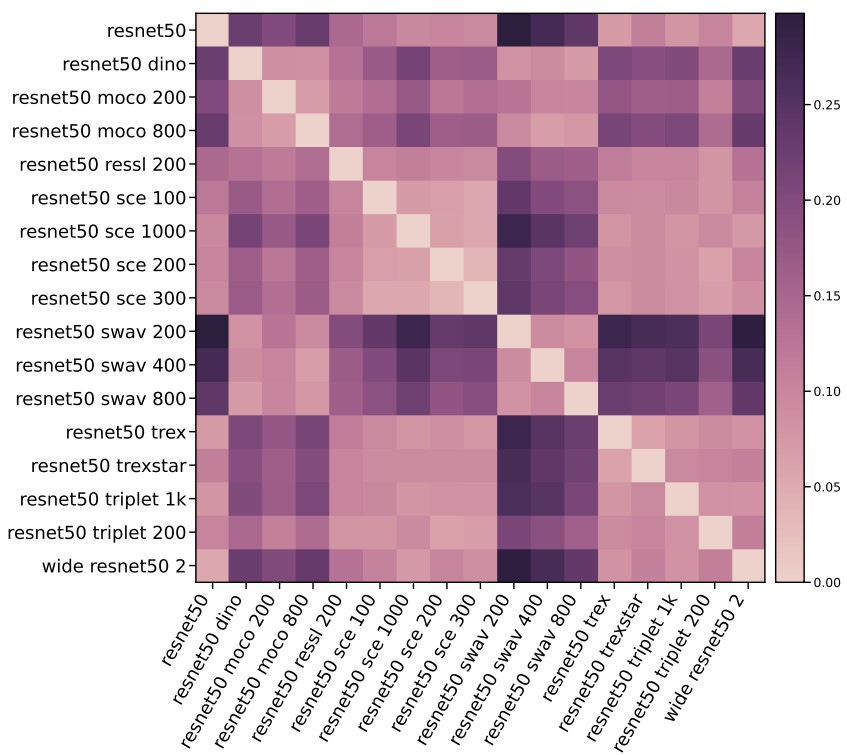

Figure 17: DTW distances associated with the clustering presented in Figure 16. We can see distinct clusters as revealed by the dendrogram.

