# OpenReview forum: "From Alexnet to Transformers: Measuring the Non-linearity of Deep Neural Networks with Affine Optimal Transport"
_NeurIPS.cc/2024/Conference — Submitted to NeurIPS 2024_

### Official Review · Reviewer_8vYn · 2024-07-04

**Soundness:** 3
**Presentation:** 4
**Contribution:** 3
**Rating:** 7
**Confidence:** 2

**Summary:**

This paper proposes the affinity score, which measures the non-linearity of an activation function $\sigma(X)$ given the distribution of $X$.
The affinity score is defined based on how well the 2-Wasserstein distance $W_2(X, Y)$, where $Y=\sigma(X)$, is approximated by $W_2(N_X, N_Y)$, where $N_X$ and $N_Y$ are Gaussian approximations of the distributions of $X$ and $Y$, respectively.
Note that $W_2(N_X, N_Y)$ has a closed-form solution, and it holds that $W_2(X, Y) = W_2(N_X, N_Y)$ if the relation between $X$ and $Y$ is locally affine on the support of the given $X$.
The authors then propose to characterize a DNN model by the set of affinity scores of activation functions in the model under a given input distribution.
Experimental results suggest that the affinity scores are relatively low in transformer-based vision models, meaning that the activation functions are used in a more non-linear region compared to CNN models.

**Strengths:**

* The proposed score presents an interesting insight in comparing the series of CNN models and transformer-based models. Experiments suggest that transformer-based models utilize the non-linearity of activation functions more efficiently, leading to the higher prediction performance.

**Weaknesses:**

* It is empirically shown that the proposed score has a low correlation with existing non-linearity metrics such as R^2, but it is unclear whether the existing metrics are insufficient to analyze the DNN models in the way proposed in this paper. I would like to see how the distribution in Fig. 3(C) changes when other metrics such as R^2 are used instead of the proposed $\rho_{aff}$.
* In my opinion, one would expect the nonlinearity score to behave symmetrically at $x=0$ for activation functions like ReLU, but the proposed affinity score seems to have a lower score at negative $x$, as shown in Fig.2 or Fig.6. Is there any reasonable explanation for such a behavior of the proposed score?

**Questions:**

* I could not fully understand the definition of the non-linearity signature in Def. 3.1. It is defined based on $F_i \cap \mathcal
{A}$, but $F_i$ is a layer in N, so it seems to be empty.

* This is just a comment out of interest, I would like to see if the affinity score of a neuron in the same model changes depending on the input class.

**Limitations:**

See weakness above.

---

> ### Author Rebuttal · Authors · 2024-08-07
>
> We thank the reviewer for taking the time to review our work and for their comments.
>
> > I would like to see how the distribution in Fig. 3(C) changes when other metrics such as R^2 are used instead of the proposed $\rho_{aff}$.
>
> We thank the reviewer for this suggestion. We've added histograms for $R^2$ score similar to those presented in Figure 3 for the affinity score in the Fig.1 of the PDF included in the global rebuttal. We can see vividly that $R^2$ score is less informative, with several models having modes around negative values. This is in line with our Remark 3.6 explaining why such a metric may be inappropriate for the considered task.
>
> > one would expect the nonlinearity score to behave symmetrically at $x=0$ for activation functions like ReLU
>
> This is an interesting question, as it is a counterintuitive behavior. We can study this behaviour analytically by restricting to $X \sim \mathcal{U}[b,a]$, $b < 0$ and $a > 0$, $f: x \mapsto \mathrm{ReLU}(x)$, $Y = f(X)$ and verifying whether $\rho_{aff}(X, Y)$ is symmetrical in $a$ and $b$, that is, if it's invariant with respect to the assignment $(a,b)\mapsto (-b, -a)$.
>
> A straightforward computation yields
> - $\mu(X) = \frac{a+b}{2}$, $\Sigma(X) = \frac{(a-b)^2}{12}$,
> - $\mu(Y) =\frac{a^2}{2(a-b)}$ and $\Sigma(Y) = \frac{a^3(a-4b)}{12(a-b)^2}$.
>
> Plugging these in the affine transport formula yields
>
> - $A_{aff}=\frac{\sqrt{a^3(a-4b)}}{(a-b)^2}$
> - $b_{aff} = \frac{a}{2(a-b)}\left(a - \sqrt{a(a-4b)}\left(\frac{a+b}{a-b}\right)\right)$.
>
> Finally, we can compute
>
> $W_2^2(T_{aff}(X), Y) = \frac{a^3}{6(a-b)^2}\left((a-4b) + \sqrt{a(a-4b)}\left(\frac{-a+3b}{a-b}\right)\right)$.
>
> Here we can see that this term is not strictly symmetric, but empirically we can assess that this term is not too far from being symmetric (see leftmost part of Figure 3 in the attached PDF).
>
> A greater source of asymmetry in the nonlinearity score
>
> $\rho_{aff}(X,Y) = 1 -  \frac{W_2(T_{aff}(X), Y)}{\sqrt{2Tr(\Sigma(Y))}}$
>
> is the trace term, as can be seen from the expression for $\Sigma(Y)$ and the middle plot of Figure 3.
> We illustrate this in the attached PDF in the global rebuttal.
>
> > I could not fully understand the definition of the non-linearity signature in Def. 3.1. It is defined based on $F_i \cap \mathcal{A}$, but $F_i$ is a layer in N, so it seems to be empty.
>
> We consider a layer $i$ as a function $F_i$ acting as a block used in a repeated way in a given architecture. In that sense, $F_i \cap \mathcal{A}$ would just be a set of activation functions in such block without all other eventual operations (such as for instance, convolutions).
>
> > Affinity score of a neuron in the same model depending on the input class.
>
> We thank the reviewer for this suggestion. We provide a visualization (Fig.4 of the attached PDF in the general reply) of the variance of the affinity scores when passing batches consisting of samples belonging to a single class of CIFAR10 dataset (so we have 10 batches, each batch contains samples from 1 class). For the sake of clarity, we use models with less than 40 layers. One can observe that early layers seem to be the most sensitive to the class distribution for all models. For ViT, some MLP block of intermediate layers also exhibit a high variability potentially hinting at their mono- and poly-semanticity.

---

> > ### Comment · Reviewer_8vYn · 2024-08-12
> >
> > Thank you for your responses.
> >
> > From Figure 1 in the supplemental pdf, it appears that R^2 has a similar tendency to $\rho_{aff}$, e.g., the bimodality of ResNet152 and the less linear behavior of Vit Huge 14. As the author notes, it is curious that R^2 becomes negative.
> >
> > Figure 4 in the supplemental pdf is interesting, showing another difference in Conv nets and ViTs. I would like the authors to include these results in the main manuscript.
> >
> > I would like to keep my score at 7.

---

> > > ### Author Response · Authors · 2024-08-13
> > > **Thank you**
> > >
> > > We thank the reviewer for their reply. We are happy that the reviewers found the provided illustrations insightful and we will include them in the new revision as suggested.

---

### Official Review · Reviewer_ctip · 2024-07-11

**Soundness:** 2
**Presentation:** 2
**Contribution:** 1
**Rating:** 4
**Confidence:** 4

**Summary:**

This study proposes empirical statistics about different DNN architectures in the hope to shed some light into why some architectures are better than others for some computer vision tasks. To do so, the study leverages common optimal transport results on DNN's internal representations, under some strong assumption about the distribution of those representations.

**Strengths:**

The paper proposes to consider an interesting and useful question of going to the bottom of why some architectures are better than others as measured by some restricted downstream task.

**Weaknesses:**

- I do not agree with the following statement `Without non-linear activation functions,
84 most of DNNs, no matter how deep, reduce to a linear function unable to learn complex patterns.` as to me, models such as transformers with linear attention and linear MLP blocks have no actual nonlinearity but are higher order polynomial of the input, i.e., are not linear. Could the authors provide clarifications on that statement or did I misunderstand something?

- I also disagree with the following `Activation functions were also early identified [29, 30, 31, 32] as a key to making even a shallow
86 network capable of approximating any function, however complex it may be, to arbitrary precision.` since again, Fourier series for example can approximate any function as well. Hence DNN nonlinearities are certainly not the key ingredient to function approximation in general

- Many formal results such as Theorem 3.3 are well known and have been established for years (even decades) but no reference is provided which is misleading to the reader.

- Fig 2. is also misleading since the "nonlinearity" of any activation function depends on the range of the inputs. The only case that wouldn't be true is e.g. for ones with constant second derivatives, i.e., a linear activation function.... hence again that statement is highly misleading in presenting ReLU as inherently benefiting form that property compared to others

- the statement `No other metric extracted from the activation functions of the
260 considered networks exhibits a strong consistent correlation with the non-linearity signature.` is again an overstatement as the authors only compare with a few alternatives and theorem is provided to support such a statement

- the statement `We proposed the first sound approach to measure non-linearity of activation functions in neural
270 networks` is also incorrect, see e.g.
  - https://jmlr.org/papers/v20/18-418.html
  - https://arxiv.org/pdf/2301.09554
  - https://arxiv.org/abs/1810.09274
  all the above works have been published in peer reviewed journals/conferences

**Questions:**

Please see the **Weaknesses** section

**Limitations:**

In addition to my concerns expressed above, the study does not provide any actionable insights or understanding on the "why" of different architectures performing differently beyond the proposed statistical numbers. How could one use the provided analysis to better design model architectures or for model selection?

Also, the paper does not provide any novel theoretical results. All the major theorems and results are already widely known within the community, yet they are presented as part of the contributions. With that in mind, the paper solely leverages existing OT tools, with some underlying simplifications on the DNN's data distribution, and report computed metrics. Hence the study falls below acceptance level in my opinion and would need a major rewriting + additional novel contributions to be worth acceptance.

The writing style is also filled with unsupported claims and highly misleading statements (see the **Weaknesses** examples).

---

> ### Author Rebuttal · Authors · 2024-08-07
>
> We thank the reviewer for taking the time to review our work and for their comments.
>
> > I do not agree with the following statement `Without non-linear activation functions, most of DNNs, no matter how deep, reduce to a linear function unable to learn complex patterns.`
>
> We believe that there is a misunderstanding regarding this claim. First, we believe that the presence of activation functions in all models available on Torchvision suggests that they are necessary to learn complex patterns. We are not aware of any DNN architecture that is widely acclaimed and that removes the activation functions altogether. Yet, we agree that for some models, such as transformers, even removing the activation function from attention and MLP blocks, doesn't make them linear due to the quadratic term in the attention equation. We propose to reformulate this phrase by saying: "For instance, without non-linear activation functions, popular multi-layer perceptrons, no matter how deep, reduce to a linear function unable to learn complex patterns."
>
> > I also disagree with the following `Activation functions were also early identified [29, 30, 31, 32] as a key to making even a shallow network capable of approximating any function, however complex it may be, to arbitrary precision.`
>
> We **do not** claim that "nonlinearity is the key ingredient to function approximation in general" as the reviewer says. Here, we refer to universal approximation theorems [29, 30, 31, 32] of feedforward networks that require at least one hidden layer and a non-linearity to be proved. If there are other universal approximation theorems that the reviewer has in mind and that can be proved for models without an activation function, we will gladly discuss them in our work and adjust the phrasing accordingly.
>
> > Many formal results such as Theorem 3.3 are well known
>
> We respectfully disagree with this comment as they seem to confuse the common results for OT between Gaussian distributions, duly cited in our work, and Theorem 3.3. Our statement holds for **arbitrary (non-gaussian)** random variables $X$ and $Y$ that satisfy $Y=T(X)$ for a symmetric positive-definite linear transformation $T$. The classical result about normal distributions is then a direct corollary: if $X$ and $Y$ are normally distributed, then such a $T$ can always be found, and the classical result follows.
>
> We also note that "Computational Optimal Transport" by Peyré and Cuturi (2019), which arguably represents a modern reference for applied OT doesn't contain any results having the generality of Theorem 3.3. Yet, it does refer to the classical result for OT between Gaussians in Remark 2.31 as well as its generalization to elliptical distributions in Remark 2.32.
>
> > Fig 2. is also misleading since the "nonlinearity" of any activation function depends on the range of the inputs.
>
> Kindly note that we illustrate exactly what the reviewer claims in Fig. 2 (A) for ReLU and in Appendix D Fig.6 for 8 more activation functions. In Appendix D, we explicitly say, "We observe that **each** activation function can be characterized by 1) the lowest values of its non-linearity obtained for some subdomain of the considered interval and 2) the width of the interval in which it maintains its non-linearity." (lines 514-516). We propose to clarify this in the caption of Fig. 2 (A) by saying "Non-linearity of activation functions depends on the range of input values (red), illustrated here on ReLU".
>
> >the statement No other metric extracted from the activation functions of the considered networks exhibits a strong consistent correlation with the non-linearity signature. is again an overstatement
>
> As explained in our work, we are not aware of any other quantifiable way to measure the non-linearity of an activation function. Yet, we considered reasonable alternatives to it: such as $R^2$ score, CKA commonly used to measure the similarity across the layers of NNs, and sparsity of activations recently used to better understand the semantic properties of neurons in MLPs and transformers. Our results show that the non-linearity signatures capture something different from these metrics, just as we claim it. We will gladly discuss this with the reviewer.
>
> > the statement We proposed the first sound approach to measure non-linearity of activation functions in neural networks is also incorrect
>
> None of the works mentioned by the reviewers provides a measure of non-linearity for activation functions. If we missed the introduction of such a metric upon reading the above-mentioned works, we would like to kindly ask the reviewer to point out the exact equations where it is defined in each of these works.
>
> > the study does not provide any actionable insights or understanding on the "why"
>
> Our experiments put forward several novel contributions factually explaining "why" different neural architectures differ from the lens of their activation functions' non-linearity. We show that DNNs of the last decade have several trends for which **we do not find any mention in prior work**: 1) Till 2020, state-of-the-art CNNs included more (on average) linear activation functions, 2) ViTs behave differently from CNNs with a much more non-linear behavior of the activation function in their MLP blocs, 3) skip connections help the model learn the (linear) identity function, as suggested by He et al. (2016).
>
> As highlighted in the Discussion section, the affinity score **does provide** actionable insights: it allows comparing neural architectures and identifying those that have a potential to disrupt the ML field, just as the transformers did.
>
> > the study leverages common optimal transport results on DNN's internal representations, under some strong assumption about the distribution of those representations.
>
> We stand by our claim expressed before and hope that our reply regarding Theorem 3.3 which holds for **arbitrary random variables** will clear out the misunderstanding that the reviewer insists on.

---

> > ### Comment · Reviewer_ctip · 2024-08-12
> > **Thank you**
> >
> > I want to thank the authors for providing clear answers to each of my concerns. In light of those findings and while considering the other reviews I have updated my score. I still believe that some improvements could be done in terms of presentation (which would have helped my initial assessment) and I am still unconvinced that the method allows for "actionable results". Also, I find that many insights are echoing previous studies and are not surprising (which is why I raised my score to 4 and not 5).

---

> > > ### Author Response · Authors · 2024-08-13
> > > **Thank you for your reply and reconsidering the score**
> > >
> > > We would like to thank the reviewer for reconsidering their score.
> > >
> > > > I find that many insights are echoing previous studies
> > >
> > > If our study echoes other known results on this topic, we would gladly discuss and contextualize them in our work if the reviewer kindly provides us with more specific pointers to such studies. There seem to be no other metrics for measuring non-linearity as we propose to do it but we remain open to discovering other works deriving similar insights with other tools. We genuinely tried to find reasonably related works for this contribution but didn't manage to find them.
> > >
> > > > I am still unconvinced that the method allows for "actionable results"
> > >
> > > As for the providing actionable results, can the reviewer elaborate a bit more on what is considered actionable? We believe that our study provides comparisons between models in a theoretically-grounded way. It is thus actionable as it allows to quantify a quantity of interest and use this quantity in a variety of ways (to compare models, to understand the role of different parts of the network etc). We believe that such a comparison can be done in future works as well, acting as a tool in them. We remain open to discussing it further until the end of the rebuttal period if the reviewer shares with us they understand by actionable.

---

### Official Review · Reviewer_2p6h · 2024-07-13

**Soundness:** 4
**Presentation:** 3
**Contribution:** 4
**Rating:** 7
**Confidence:** 4

**Summary:**

This paper introduces a novel method for quantifying the non-linearity of activation functions in neural networks, termed the "non-linearity signature." Using an affinity score derived from optimal transport theory, it measures the non-linearity of individual activation functions. It defines the non-linearity signature as a comprehensive set of these scores across all functions in a deep neural network (DNN). The study compares these signatures across a range of popular DNN architectures in computer vision, revealing clear patterns in their evolution over the past decade, notably showing a trend towards decreasing non-linearity until the disruptive impact of vision transformers. It emphasizes the uniqueness of their measure, as it does not strongly correlate with other metrics across different architectures. The approach could potentially be applied to analyze the non-linearity of newer large language models (LLMs) and identify innovative neural architectures that optimize internal non-linear characteristics for enhanced performance, crucial in the era of costly experiments with large-scale model optimizations.

**Strengths:**

1. Novelty and importance. The paper introduces a theoretically grounded measure, the affinity score, for quantifying the non-linearity of activation functions using optimal transport theory, providing a robust framework for analysis. This is the first approach to approximately measure the non-linearity of DNNs, which is crucial for understanding their inner workings.
2. Solid theoretical and experimental validation.  The method is grounded in optimal transport theory, providing a rigorous theoretical foundation for the proposed non-linearity signature. This enhances the credibility and robustness of the findings. The experimental results demonstrate the practical utility of the non-linearity signature. It can predict DNN performance and meaningfully identify the family of approaches to which a given DNN belongs, making it a valuable tool for researchers and practitioners.
3. Clear Writing. The structure of the paper is well-organized, with a clear presentation of background knowledge, theoretical properties, experimental evaluations, and conclusions. This clarity aids in understanding the contributions and implications of the research. The paper's figures and tables are comprehensive, providing clear and precise information, and the writing maintains a coherent logical sequence.

**Weaknesses:**

1. The authors should discuss more activation functions. Currently, only ReLU, Tanh, and Sigmoid are included.  While these are among the most commonly used activation functions in neural networks, many other activation functions have been introduced and shown to be effective in various contexts, like GELU. Including a more comprehensive analysis of a diverse set of activation functions would enhance the robustness and applicability of their proposed method.
2. There is currently some research on the nonlinearity of deep neural networks that should be compared and discussed.
3.  It would be beneficial to showcase examples from domains beyond computer vision. While the paper focuses on computer vision tasks, it may not address the non-linearity signature's applicability to other domains such as NLP, speech recognition, or reinforcement learning. The findings might be less generalizable if the proposed measure does not perform equally well across diverse types of tasks and data.

**Questions:**

How would the model’s nonlinearity affect different downstream tasks? Does the impact vary across datasets?

**Limitations:**

Yes, they have discussed the assumption of Theorem 3.3.

---

> ### Author Rebuttal · Authors · 2024-08-07
>
> We thank the reviewer for their positive comments about our work.
>
> > The authors should discuss more activation functions
>
> We thank the reviewer for this suggestion. We kindly note that the individual behavior of 9 different activation functions (Sigmoid, ReLU, GeLU, ReLU6, LeakyReLU, tanh, hardtanh, SiLU and Hardswish) is shown in Fig. 6 in Appendix D. Additionally, we also consider the original activation functions used for each neural architecture studied further in our work (GeLU for ViTs, Swin, ConvNext, and SiLU in EfficientNets).
> > There is currently some research on the nonlinearity of deep neural networks that should be compared and discussed.
>
> There was a recent paper by Teney et al. ("Neural Redshift: Random Networks are not Random Functions", CVPR, **June 2024**) that discusses the biases of the different activation functions for randomly initialized DNNs. Although it **doesn't introduce a non-linearity measure** for activation functions, it does illustrate the inductive bias of varying activation functions and their sensitivity to a change of weight magnitudes in Glorot initialization for randomly initialized MLPs. The authors of the latter work conclude that different activation functions visually carry different inductive biases (low complexity for ReLU, higher complexity for Tanh) that are more or less sensitive to weight magnitudes. Our work allows quantifying their findings using the affinity score (panel (a) of Fig.2 in the attached PDF). We also extend their work by showing that these biases can be changed by shifting the domain of the weight initialization (toward negative values) following the intuition provided in our work (panel (b) of Fig.2).  This also affects the affinity score that becomes lower confirming our findings from the main manuscript.
>
> We are open to discussing any other relevant work we may have missed further.
>
> > It would be beneficial to showcase examples from domains beyond computer vision.
>
> We thank the reviewer for this suggestion. We considered the computer vision task, as the torchvision repository allows for a reproducible comparison of state-of-the-art models trained on Imagenet dataset. Additionally, the available models are very varied and span a decade of research. We agree that extending our analysis to other tasks mentioned by the reviewer is also very important, and we believe that our work can spur such contributions in the future. We will mention this in the Discussion.
>
> > How would the model’s nonlinearity affect different downstream tasks? Does the impact vary across datasets?
>
> We show in Appendix H and Fig. 15 the deviation of non-linearity signatures for the same model between ImageNet and different datasets (CIFAR10/CIFAR100/Random data). We can see that deviations between ImageNet and CIFAR10/CIFAR100 are much lower than for random data, implying that these datasets are processed similarly to Imagenet. Measuring the distance between the non-linearity signature of the same DNN for different datasets thus could be a promising way to measure semantic similarity between datasets with potential applications in meta- and transfer learning where this notion is of primary importance.

---

### Author Rebuttal · Authors · 2024-08-07

We thank the reviewers for their valuable feedback. We are glad to know that they found our work **novel** (Reviewer 2p6h), **insightful** (Reviewer 8vYn), our experiments **solid** (Reviewer 2p6h), and the writing **clear** (Reviewer 2p6h). Below, we summarize the additions that we present in the attached PDF following reviewers' questions.

1. We provide a comparison to a very recently published work of Teney et al. (CVPR'24): "Neural Redshift: Random Networks are not Random Functions" showing how the neural redshift phenomenon can be understood through the lens of the affinity score, **although it doesn't introduce a non-linearity measure** contrary to our work. The latter paper illustrates the inductive bias of different activation functions by passing an image formed by a 2D grid with values distributed uniformly in $[-1,1]^2$ interval and their sensitivity to a change of weight magnitudes in Glorot initialization for randomly initialized MLPs. The authors of the latter work then conclude that different activation functions visually carry different inductive biases (low complexity for ReLU, higher complexity for Tanh) that are more or less sensitive to weigh magnitudes (panel (a) of Fig.2). We quantify them with our affinity score and confirm quantitatively their findings. We also extend their work by showing that these biases can be changed by shifting the domain of the weight initialization (toward negative values) following the intuition provided in our work (panel (b) of Fig.2).  This also affects the affinity score that becomes lower confirming our findings from the main manuscript.
2. Following the request of Reviewer 8vYn, we've added histograms for $R^2$ score in Fig.1 similar to those presented in Figure 3 for the affinity score. We can see vividly that $R^2$ score is less informative with several models having modes around negative values. This is in line with our Remark 3.6.
3. We added a plot (Fig.3) showing the evolution of the two terms defining the affinity score to explain its asymmetry. The idea behind it is to illustrate how the two evolve in a simple 1d case and when the affinity score approaches 0. A more detailed analytical analysis for this is given in the response to Reviewer 8vYn.
4. Following the request of Reviewer 8vYn, we added a plot (Fig.4 in the attached PDF) showing how the affinity scores vary in the different layers of several DNNs when samples from different classes of CIFAR10 dataset are passed through it. One can observe that early layers seem to be the most sensitive to the class distribution for all models. For ViT some MLP block of intermediate layers also exhibit a high variability.

We hope that these additional experiments and the explanations provided to reviewers in their individual responses address their questions. We respectfully encourage the reviewers to endorse our paper if our replies are satisfactory to them.

---

### Decision · Program_Chairs · 2024-09-25

**Decision:**

Reject

**Comment:**

This paper introduces a method to quantify the non-linearity of individual activation functions in neural networks using common optimal transport results. It reveals distinct patterns in the evolution of non-linearity across vision architectures. The reviewers have raised various valid concerns about the writing or the applications of this paper. In addition, concerns about the novelty of the theoretical results have been mentioned, but those have been of secondary nature in the decision. The writing along with the insights should be improved, along with the rigor of the theoretical results.